# MTG-RPD: Multimodal Trajectories Generation with Rule-Based Prior Diffusion for End-to-End Autonomous Driving

## Abstract

Replicating human driving behaviors in complex and authentic real-world environments remains a key challenge in autonomous driving. While end-to-end autonomous driving technologies have advanced substantially, generating safe and diverse multimodal trajectories poses a persistent hurdle. In recent years, diffusion-based methods have demonstrated remarkable potential across image generation, robotics, and autonomous driving—with trajectory generation approaches based on diffusion models also emerging. However, balancing real-time performance and reconstruction accuracy remains an unresolved issue. To address these limitations, we propose MTG-RPD, an innovative trajectory generation method that integrates rule-based prior knowledge. The approach first generates trajectory anchor points via rule-based prior clustering, then leverages a conditional diffusion model to transform an anchored Gaussian distribution into a multimodal trajectory distribution under scene information guidance. Notably, the diffusion model is specifically designed to facilitate agent-agent and agent-environment interactions. On the planning-based NAVSIM dataset, MTG-RPD achieved a PDMS of 88.5 when evaluated using the ResNet-34 backbone network. Code at `https://anonymous.4open.science/r/test1_try/README.md`

## 1 Introduction

Autonomous driving technology has garnered widespread attention due to its remarkable performance in enhancing traffic safety and driving efficiency. Early autonomous driving systems typically adopted a sequential process encompassing perception, prediction, and planning, which gave rise to issues such as inconsistent optimization objectives and information loss across module boundaries. Since the advent of UniAD (Hu et al., 2023), end-to-end technology has become increasingly prevalent in autonomous driving systems. In this paradigm, intermediate tasks like mapping and detection no longer function as independent components but serve as auxiliary processes to support the core planning task. End-to-end autonomous driving technology aims to take raw sensor data (e.g., multi-view images, LiDAR point clouds, and ego-vehicle states) as input and directly output the desired trajectory to guide future motion planning. Compared with traditional rule-based motion planning, end-to-end motion planning offers higher scalability, as it can be extended to more complex driving scenarios.

To better learn driving behavior patterns from data, early methods, such as Transfuser (Chitta et al., 2022), UniAD, and VAD (Jiang et al., 2023a), generating a single reasonable trajectory by imitating the driving behavior of human experts in the training dataset. However, these methods have obvious limitations: they neither account for the inherent uncertainty and multimodal characteristics of driving behavior nor adapt well to out-of-distribution scenarios, which severely restricts their practical deployment in real-world environments. Consequently, recent research has increasingly shifted toward multimodal trajectory generation, with current mainstream approaches falling into two categories.

The first category is represented by the Hydra-MDP series (Li et al., 2024; 2025b;d). Its core approach simplifies the problem of generating high-quality multimodal trajectories in a continuous

action space into selecting the optimal candidate trajectory from a fixed planning vocabulary. By discretizing the action space into a finite set of predefined motion primitives, these methods achieve efficient multimodal output but may compromise trajectory smoothness and fail to cover all potential reasonable behaviors due to the fixed nature of the vocabulary.

The second category is represented by DiffusionDrive (Liao et al., 2025), GoalFlow (Xing et al., 2025), and TransDiffuser (Jiang et al., 2025), which aim to extend the successful application of diffusion models (Ho et al., 2020) to the robotics domain (Jiang et al., 2023b; Zhu et al., 2024) to the multi-modal trajectory generation task in end-to-end autonomous driving. Specifically, it uses scene and motion information as conditional inputs to generate candidate multi-modal trajectories. In practice, GoalFlow imposes strong constraints on the trajectory generation process by constructing a dense vocabulary of anchor trajectory endpoints; whereas DiffusionDrive and TransDiffuser focus on the issue of pattern collapse — due to different random noise inputs converging to similar trajectories during denoising, resulting in a lack of diversity in generated trajectories.

However, despite the progress made by diffusion-based methods in multimodal trajectory generation, challenges remain in balancing real-time performance and reconstruction accuracy, especially in complex scenarios involving dynamic interactions between agents and the environment. These issues hinder their reliable deployment in real-world autonomous driving systems. To address these limitations, this paper proposes MTG-RPD, a novel multimodal trajectory generation method that integrates rule-based prior knowledge into the diffusion framework. Specifically, MTG-RPD first generates trajectory anchor points through rule-based prior clustering, providing a structured foundation for subsequent trajectory generation. It then employs a conditional diffusion model to transform an anchored Gaussian distribution into a multimodal trajectory distribution under the guidance of scene information, with the diffusion model explicitly designed to capture and model agent-agent and agent-environment interactions, thereby enhancing the adaptability of generated trajectories to complex scenarios.

Our contributions can be summarized as follows:

1) Integrating rule-based prior knowledge into the diffusion model to generate multimodal trajectories and improve trajectory planning real-time performance.

2) Designing a diffusion model that supports interactions between agents and between agents and the environment.

3) Achieving a PDMS of 88.5 on the NAVSIM dataset, improving trajectory generation performance.

## 2 RELATED WORK

### 2.1 END-TO-END AUTONOMOUS DRIVING

End-to-end autonomous driving systems have been at the forefront of research in recent years, primarily owing to their potential to integrate perception, planning, and control into a single unified framework. Early research, exemplified by UniAD, pioneered the end-to-end autonomous driving paradigm. By integrating multi-perception tasks—including object detection, semantic segmentation, and motion prediction—through joint training, this model achieved perception-planning integration. Its core technology lies in a single-trajectory autoregressive generation mechanism: based on historical state sequences, it progressively predicts future trajectory points step-by-step using temporal models such as GRU, ultimately outputting a single deterministic path. Subsequent studies such as Transfuser refined this paradigm by leveraging Transformers to fuse multimodal sensor data and enhance environmental representation capabilities, while PARA-Drive (Weng et al., 2024) significantly improved system real-time performance through a parallelized architecture that concurrently executes tasks like mapping and occupancy prediction. The core advantages of this paradigm include a lightweight structure and efficient inference; however, constrained by its sequential point-wise generation mechanism, it struggles with multiple-solution scenarios in complex interactive environments.

To address the limitations of single-modality planning in uncertain scenarios, researchers have shifted to multimodal trajectory planning, whose essence involves generating multiple candidate trajectories and selecting the optimal solution through probabilistic or rule-based screening. VADv2

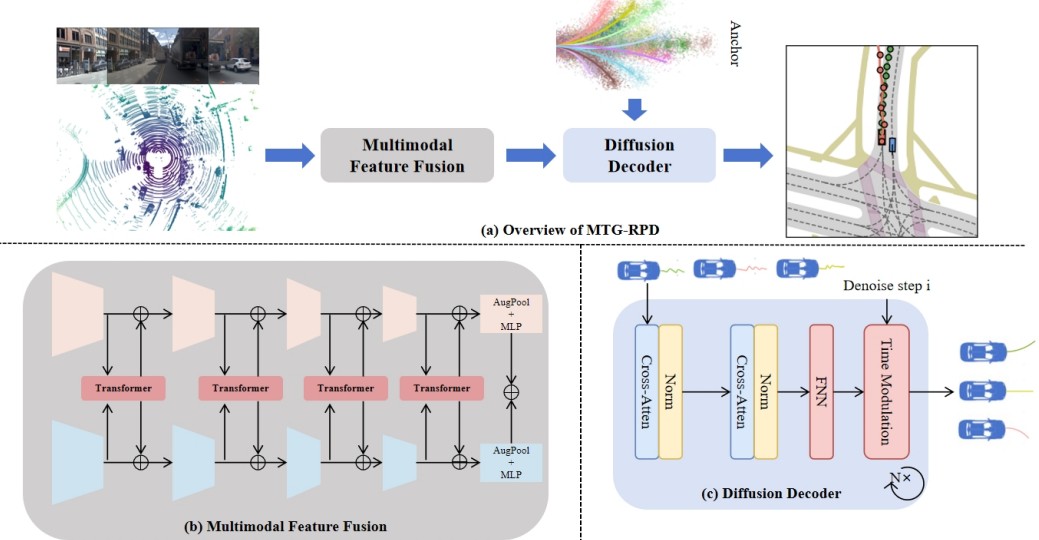

Figure 1: Overview of MTG-RPD. (a) MTG-RPD consists of two modules: (b) multimodal feature fusion and (c) diffusion decoder. MTG-RPD takes forward-view images and point clouds as input and outputs a sequence of trajectory points.

(Chen et al., 2024) first proposed the anchor trajectory vocabulary paradigm: predefined hundreds of trajectory templates covering typical driving behaviors, where the model scores each trajectory via semantic encoding and samples the highest-probability path. This design explicitly decouples trajectory generation and scoring, significantly enhancing planning diversity. The Hydra-MDP series further optimized the scoring mechanism and incorporated a rule-based expert scorer to provide supervision signals. Recently emerged diffusion models have broken traditional sampling limitations: DiffusionDrive proposed a truncated diffusion policy, modeling trajectory generation as a denoising process from anchored Gaussian distributions toward target paths. Compared to VADv2's discrete anchor sampling, this technique leverages continuous probability modeling to maintain trajectory diversity. TrajHF (Li et al., 2025a) introduced reinforcement learning fine-tuning within the diffusion framework, optimizing long-tail performance such as emergency obstacle avoidance using proprietary scenario data.

## 2.2 DIFFUSION FOR AUTONOMOUS DRIVING TRAJECTORY PLANNING

Generative models have revolutionized data synthesis across domains. Generative Adversarial Networks (Goodfellow et al., 2014) and Variational Autoencoders (Kingma et al., 2013) pioneered image generation by learning data distributions through adversarial training or variational inference. Diffusion models, emerging as state-of-the-art generative frameworks, iteratively denoise random noise into structured outputs via Markov chains. Their hierarchical denoising process enables high-fidelity synthesis in vision (Zhao et al., 2023; Po et al., 2024) and language tasks (He et al., 2022; Nie et al., 2025), surpassing earlier models in capturing complex multimodal distributions.

Recent end-to-end autonomous driving research has adopted diffusion models for trajectory planning. Unlike traditional single-path approaches, diffusion planners generate multimodal candidate trajectories by denoising Gaussian noise into plausible paths. DiffusionDrive proposes a truncated diffusion strategy that begins denoising from an anchored Gaussian distribution (rather than a standard Gaussian distribution), thereby reducing the number of denoising steps while maintaining diversity. Meanwhile, GoalFlow uses a goal point vocabulary to guide denoising and mitigate mode collapse. These innovations demonstrate that diffusion models outperform autoregressive or scoring-based methods in generating diverse, human-like trajectories.

## 3 METHOD

### 3.1 PRELIMINARY

**Task description.** This paper addresses the task of end-to-end autonomous driving, where the system directly generates the ego vehicle's future trajectory from raw sensor inputs through a neural network model. The inputs encompass multi-view camera images $C$, LiDAR point clouds $L$, and the ego vehicle's state parameters $S$. The output is a planned trajectory, represented as a sequence of future waypoints $\tau = \{(x_t, y_t)\}_{t=1}^{T_f}$, where $T_f$ denotes the planning horizon and each $(x_t, y_t)$ specifies a coordinate in the ego vehicle's frame. This task can be formally defined by the following mapping function:

$$\tau = f(C, L, S) \tag{1}$$

**Overall Architecture.** The proposed MTG-RPD framework employs an encoder-decoder architecture that integrates a perception feature fusion module and a trajectory generation module to achieve end-to-end trajectory planning for autonomous driving. The encoder extracts and fuses features from visual and point cloud data, while the decoder utilizes a diffusion model to generate trajectory plans. A key innovation of the framework lies in incorporating rule-based prior trajectories into the diffusion process, enabling efficient generation of safe and diverse multimodal trajectories while maintaining real-time performance and reconstruction accuracy.

The architecture commences with a *Multimodal feature fusion*, which processes multi-source sensor inputs to derive comprehensive environmental and agent-specific features. Raw data from multi-view images $C$, LiDAR point clouds $L$, and ego-vehicle state $S$ are fed into this module: images and point clouds undergo independent feature extraction via ResNet networks, followed by cross-modal fusion using multi-resolution Transformer modules to generate BEV scene features $F_{bev}$; concurrently, the ego-vehicle state $S$ is encoded through an MLP, and combined with $F_{bev}$ via a TransformerDecoder (guided by a learnable query $Q$) to produce agent features $F_{agent}$. These fused features $F_{bev}$ and $F_{agent}$ serve as contextual conditions for the subsequent trajectory generation process.

Subsequently, the *diffusion trajectory generation module* takes $F_{bev}$ and $F_{agent}$ as joint inputs, leveraging a multi-modal anchor prior mechanism to drive trajectory synthesis. Anchored Gaussian distributions, precomputed via K-Means clustering on rule-based trajectory priors, initialize the diffusion process, reducing the denoising steps to merely 2 iterations for enhanced real-time performance. A cascaded diffusion decoder within this module facilitates dynamic interactions between trajectory features, environmental cues, and agent intentions through sparse deformable attention and cross-attention mechanisms, ensuring generated trajectories comply with physical constraints and scene logic. Finally, the output layer selects the highest-confidence trajectory from the multi-modal candidates generated by the diffusion module, yielding the final waypoint sequence $\tau$. This end-to-end architecture effectively bridges perception and planning, with the rule-based prior anchoring mechanism addressing the trade-off between efficiency and accuracy in diffusion-based trajectory generation.

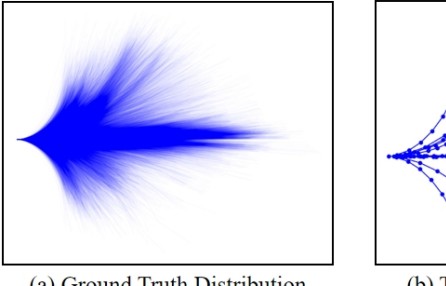

(a) Ground Truth Distribution          (b) Trajectory after K-Means

Figure 2: Trajectory distribution. (a) represents the distribution of ground truth trajectories, (b) represents trajectories after only performing K-Means

### 3.2 RULE-BASED PRIOR TRAJECTORY ANCHOR GENERATION

Real-world driving demonstrations inherently encapsulate expert knowledge and implicitly adhere to complex traffic regulations. To effectively leverage these rule priors for trajectory planning, we distill a diverse set of trajectory anchors by clustering the large-scale NuPlan dataset. These anchors serve as a robust, data-driven inductive bias within our diffusion-based planning framework, steering the generative process toward trajectories that are not only dynamically feasible but also naturally compliant with traffic rules.

However, this purely data-driven approach introduces a critical flaw: since the trajectory anchors are generated by clustering real-world trajectories, their spatial distribution inevitably inherits the inherent biases present in actual traffic flows. Taking the Nuplan dataset as an example, the dense concentration of driving trajectories on arterial roads results in a corresponding over-representation of anchors in these areas. Conversely, regions with lighter traffic but potentially higher complexity suffer from insufficient coverage, as shown in Figure 2. This severe distributional imbalance significantly undermines the model's generalization capability in long-tail scenarios. Consequently, it leads to a fundamental resource allocation paradox: the model wastes considerable computational capacity on common scenarios due to anchor redundancy, while simultaneously being exposed to critical decision-making blind spots in rare yet safety-critical situations due to the lack of prior guidance.

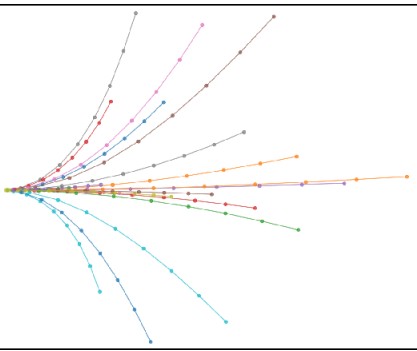

Figure 3: Trajectory anchor based on rule-based priors knowledge.

To rectify this distributional skew and construct a more balanced anchor system, we introduce a spatial rebalancing algorithm applied to the dataset prior to clustering. The core objective is to mitigate the over-concentration of trajectories in high-traffic areas without losing the diversity of driving behaviors they represent. To achieve this, we first discretize the map into a fine-grained spatial grid and compute the trajectory density within each cell. For cells where the density surpasses a predefined threshold, indicating a high-traffic zone, we perform a targeted downsampling. Instead of uniform random pruning, which could inadvertently eliminate unique maneuvers, we apply an intra-cell clustering to the resident trajectories. By retaining only the cluster centroids from these dense cells, we effectively preserve the full spectrum of driving behaviors (e.g., lane changes, turns, braking patterns) while drastically reducing their statistical dominance. This process re-weights the dataset, elevating the significance of trajectories in less-frequented regions. The subsequent clustering on this rebalanced data yields a set of anchors that provides far more uniform coverage across both common and long-tail scenarios, as shown in 3, directly resolving the resource allocation paradox and equipping the planner with a more robust and comprehensive set of priors.

### 3.3 MULTIMODAL FEATURE FUSION

To achieve robust and comprehensive environmental perception, this study adopts a multi-modal perception and fusion framework similar to that of the Transfuser model, designed to effectively integrate heterogeneous data from images and LiDAR point clouds. Compared to single-modal perception solutions, this multi-modal approach fully leverages the complementary strengths of different sensors. This complementarity is crucial for building an accurate and reliable understanding of the scene. In this multi-modal fusion module, we deploy independent Residual Networks ResNet

as feature extraction backbones for the image and point cloud data pathways, respectively. The core of the feature fusion mechanism is the introduction of multiple Transformer modules. Through their attention mechanism, these modules facilitate deep, cross-modal interaction at various feature resolution levels. Specifically, the Transformer can compute dependencies between image and point cloud features on a global scale, thereby effectively associating the fine-grained semantic information from images with the precise spatial localization from point clouds. This strategy of deep fusion across multiple scales plays a decisive role in accurately performing critical driving tasks such as obstacle localization and lane structure inference.

Following feature fusion, the Bird's-Eye View (BEV) feature of the scene, denoted as $F_{bev}$, is obtained. This BEV feature is concatenated with the ego's state features $F_{ego}$ processed by a multi-layer perceptron (MLP), and the agent feature $F_{agent}$ is derived via a learnable query $Q \in \mathbb{R}^{N_{agent} \times E}$ through TransformerDecoder. $F_{bev}$ and $F_{agent}$ serve as conditional inputs to the diffusion trajectory generation module.

$$\mathbf{F}_{\text{concat}} = \text{Concat}\left(\mathbf{F}_{\text{bev}}, \mathbf{F}_{\text{ego}}\right) \tag{2}$$

$$\mathbf{F}_{\text{agent}} = \text{TransformerDecoder}\left(\mathbf{Q}, \mathbf{F}_{\text{concat}}\right) \tag{3}$$

### 3.4 DIFFUSION DECODER

The diffusion trajectory generation module takes the fused scene BEV features $F_{bev}$ and agent features $F_{agent}$ as joint conditions, achieving efficient iterative denoising via a diffusion strategy to generate multi-modal planning trajectories that comply with physical constraints and scene interaction logic. This module employs a multi-modal anchor prior mechanism, originating from the anchored Gaussian distribution obtained through K-Means clustering, and is capable of generating high-quality trajectories with merely 1-3 denoising steps.

Specifically, the diffusion decoder serves a core function in the denoising process: first, it enables cross-modal interaction between trajectory features and BEV environmental features via a sparse deformable attention mechanism, thereby accurately capturing key information such as obstacle positions and lane structures; subsequently, it executes cross-attention operations between trajectory features and agent queries derived from the perception module, integrates the Timestep Modulation Layer to encode diffusion process information, and ultimately predicts trajectory coordinate offsets and confidence scores through an MLP. This multi-level interaction mechanism allows the model to deepen its understanding of the scene at each denoising step. For instance, in complex intersection scenarios, it can simultaneously generate diverse and reasonable trajectories such as following and lane changing.

During training, the diffusion decoder $f_\theta$ takes as input $N_{\text{anchor}}$ noisy trajectories $\{\tau_k^i\}_{k=1}^{N_{\text{anchor}}}$ and the conditional information $F_{bev}, F_{agent}$, predicts classification scores $\{\hat{s}_k\}_{k=1}^{N_{\text{anchor}}}$ and denoised trajectories $\{\hat{\tau}_k\}_{k=1}^{N_{\text{anchor}}}$:

$$\{\hat{s}_k, \hat{\tau}_k\}_{k=1}^{N_{\text{anchor}}} = f_\theta(\{\tau_k^i\}_{k=1}^{N_{\text{anchor}}}, F_{bev}, F_{agent}), \tag{4}$$

We adopt a loss calculation method similar to DiffusionDrive, using the noise trajectory corresponding to the anchor trajectory closest to the ground truth trajectory as the positive sample($y_k = 1$), and the remaining trajectories as negative samples($y_k = 1$). The training objective combines classification loss, trajectory reconstruction loss, and ego progress loss.The trajectory loss can be expressed as:

$$\mathcal{L} = \sum_{k=1}^{N_{\text{anchor}}} \left[ y_k \mathcal{L}_{\text{rec}}(\hat{\tau}_k, \tau_{\text{gt}}) + \lambda \text{BCE}(\hat{s}_k, y_k) \right], \tag{5}$$

where $\lambda$ balances the simple L1 reconstruction loss $\mathcal{L}_{\text{rec}}$ and binary cross-entropy (BCE) classification loss.

In the training phase, training samples are constructed by adding Gaussian noise to anchor trajectories, and model parameters are jointly optimized using trajectory loss, agent detection loss, and semantic map loss. During inference, trajectory samples are dynamically adjusted based on the DDIM update rule, and the trajectory with the highest confidence is ultimately selected as the output.

Table 1: Comparisons with SOTA methods in PDM score metrics on Navtest.

| Method | Modality | NC↑ | DAC↑ | EP↑ | TTC↑ | C↑ | PDMS↑ |
|---|---|---|---|---|---|---|---|
| Constant Velocity | - | 68.0 | 57.8 | 19.4 | 50.0 | 100.0 | 20.6 |
| Ego Status MLP | - | 93.0 | 77.3 | 62.8 | 83.6 | 100.0 | 65.6 |
| Transfuser | V+L | 97.8 | 92.6 | 78.9 | 92.9 | 100.0 | 83.9 |
| UniAD | V+L | 97.8 | 91.9 | 78.8 | 92.9 | 100.0 | 83.4 |
| PARA-Drive | V+L | 97.9 | 92.4 | 79.3 | 93.0 | 99.8 | 84.0 |
| DiffusionDrive | V+L | 98.2 | 96.2 | 82.2 | 94.7 | 100.0 | 88.1 |
| GoalFlow(ResNet34) | V+L | 98.3 | 93.8 | 79.8 | 94.3 | 100.0 | 85.7 |
| Hydra-MDP++(ResNet34) | V* | 97.6 | 96.0 | 80.4 | 93.1 | 100.0 | 86.6 |
| WOTE | V+L | **98.5** | 95.8 | 80.9 | 94.4 | 99.9 | 87.1 |
| VADv2 | V+L | 97.2 | 89.1 | 76.0 | 91.6 | 100.0 | 80.9 |
| **MTG-RPD (Ours)** | V+L | 98.3 | **96.7** | **82.8** | **94.8** | **100.0** | **88.5** |

# 4 EXPERIMENT

## 4.1 EXPERIMENTAL SETUP

**Dataset.** In our experiments, we utilize the planning-oriented NAVSIM dataset (Dauner et al., 2024), which is built upon the OpenScene dataset (Peng et al., 2023), a compact redistribution of nuPlan (Caesar et al., 2021), the largest publicly available annotated driving dataset, and sampled at 2 Hz. OpenScene contains 120 hours of autonomous driving data, and NAVSIM, as its end-to-end environment, leverages non-reactive simulation and closed-loop evaluations. Each sample in NAVSIM is rich in data, including camera images from 8 perspectives that achieve a full 360° field of view (FOV), fused LiDAR data from 5 sensors, ego status, and annotations for the map and objects, with annotations provided at a frequency of 2 Hz.

The dataset consists of two parts: Navtrain and Navtest, which respectively contain 1192 and 136 scenarios, totaling over 100,000 samples. In our experiments, we only use the training split (Navtrain) for training and validation to guide model selection.

**Metrics.** On the NAVSIM dataset, we evaluate the model using non-reactive simulation and closed-loop Predictive Driving Model score (PDMS) (Dauner et al., 2023). This evaluation metric quantifies driving performance by aggregating multiple sub-objectives, and its formula can be expressed as:

$$PDM_{score} = NC \times DAC \times TTC \times \frac{(5 \times DDC + 2 \times C + 5 \times EP)}{12} \tag{6}$$

The PDMS integrates two key components: penalty terms and a weighted average of subscores. The penalty terms include No at-fault Collisions (NC), Drivable Area Compliance (DAC), and Time-to-Collision (TTC) which penalize inadmissible behaviors (e.g., collisions or driving off-road) by reducing the overall score. The weighted average incorporates Driving Direction Compliance(DDC), Ego Progress (EP) and Comfort (C), with weights set to 5, 5, and 2 respectively. However, due to limitations of the NAVSIM toolkit, DDC is not calculated.

**Implementation Details.** Our code is based on the NAVSIM toolkit. In the multimodal feature fusion section, we use the same fusion method and ResNet-34 backbone network as Transfuser for a fair comparison. In the diffusion decoder, we adopt a cross-attention mechanism to interact trajectory features with agent and environment features multiple times, denoise to generate multimodal trajectories, and select the highest-scoring trajectory for evaluation. The training and inference processes are similar to Transfuser. First, three forward-view images are cropped and concatenated to form a $1024 \times 256$ image, which is input into the model along with the BEV point cloud and driving state information. The output includes the coordinates of eight path points within 4 seconds. MTG-RPD is trained from scratch on the Navtrain dataset for 100 epochs with a batch size of 256 and a learning rate of $6 \times 10^{-4}$. The model performance is evaluated on the Navtest dataset.

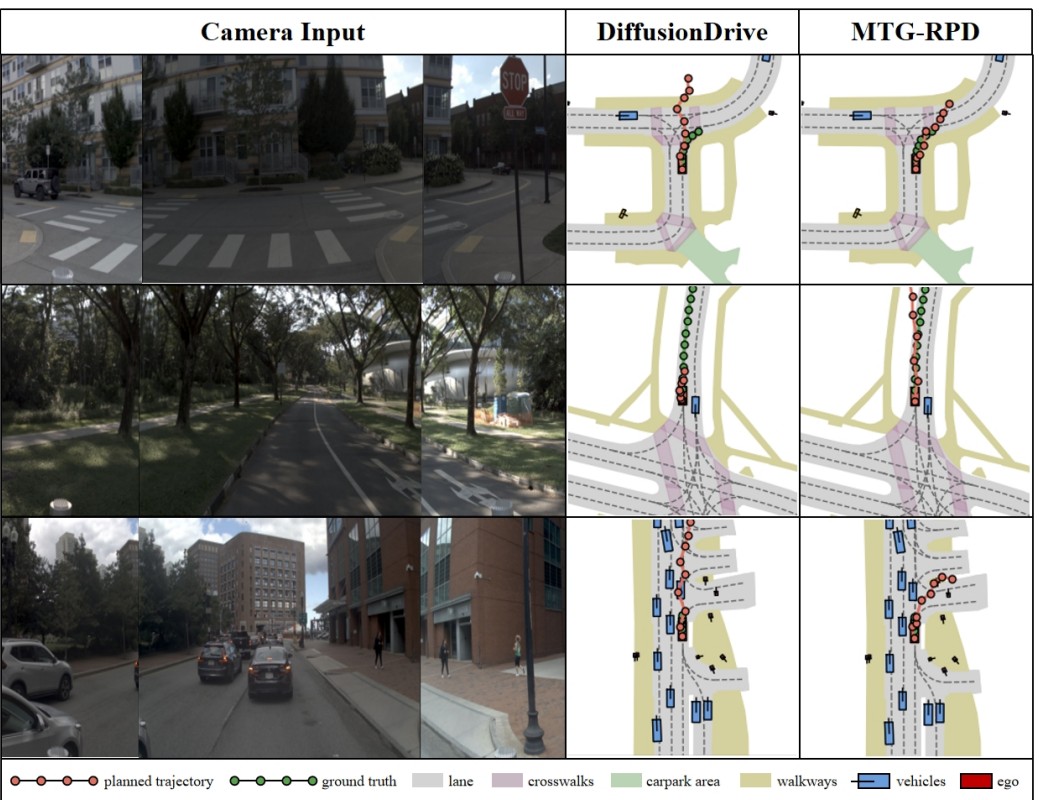

Figure 4: Qualitative comparison of our MTG-RPD model and the DiffusionDrive model. Includes intersections, smooth traffic conditions, and congested traffic conditions. The first column shows the image input, the second column shows the trajectory planning results of DiffusionDrive, and the third column shows the planning results of our MTG-RPD model.

## 4.2 QUANTITATIVE RESULTS

The quantitative results on the Navtest dataset are presented in Table 1, where our MTG-RPD model demonstrates state-of-the-art performance across several evaluation metrics. To ensure a fair comparison, ResNet-34 is consistently used as the feature extraction module for both image and point cloud data in the compared methods. Specifically, MTG-RPD achieves top-tier performance in the DAC, EP, TTC, and PDMS metrics. These results indicate that our model maintains high safety across diverse driving scenarios.

**Baseline.** In the Navtest evaluation, we compare against the following baseline methods:

**Constant Velocity:** This model assumes constant speed while moving forward, yielding limited performance as it does not account for environmental or dynamic factors.

**Ego Status MLP:** This model uses only the current state as input, leading to better performance than the Constant Velocity model but still underperforming compared to more advanced models.

**Transfuser(Chitta et al., 2022):** This method combines image and LiDAR inputs through a transformer to generate a BEV feature, which is then used for trajectory generation.

**UniAD(Hu et al., 2023):** Utilizes multiple transformer architectures to process information differently, employing queries to transfer information for planning tasks.

**PARA-Drive(Weng et al., 2024):** Differs from UniAD by performing mapping, planning, motion prediction, and occupancy prediction tasks in parallel based on the BEV feature. Notably, none of these three methods account for the multimodality of trajectories, which limits their performance.

**DiffusionDrive(Liao et al., 2025):** Employs diffusion models for trajectory prediction and planning, leveraging truncated diffusion methods to generate diverse and realistic trajectories.

**GoalFlow (ResNet34)(Xing et al., 2025):** This method uses a ResNet-34 backbone to extract features and a goal-oriented flow model to predict and plan trajectories with goal point guidance.

**Hydra-MDP++(Li et al., 2025b):** Based on a teacher-student knowledge distillation framework, this model learns from both human and rule-based teacher models to tackle multi-objective and multi-modal tasks in autonomous driving.

**WoTE(Li et al., 2025c):** This method applies a BEV world model for effective trajectory planning.

**VADv2(Chen et al., 2024):** An end-to-end driving model based on probabilistic planning, which takes sensor data as input and outputs a probability distribution over possible actions. Although these methods can generate multimodal trajectories, they still exhibit limitations in terms of overall performance.

In conclusion, the quantitative results on the Navtest dataset reinforce the superior performance of our MTG-RPD model across multiple metrics, underscoring the effectiveness of our approach for end-to-end autonomous driving tasks.To evaluate the contribution of the homogenized rule-based prior trajectory anchors in MTG-RPD, we conducted an ablation study, as summarized in Table 2.

Table 2: Results of the Ablation Study on MTG-RPD

| Method | NC | DAC | EP | TTC | C | PDMS |
|---|---|---|---|---|---|---|
| W/o Prior | 98.0 | 95.6 | 81.7 | 93.7 | 100.0 | 87.1 |
| MTG-RPD | 98.3 | **96.7** | **82.8** | **94.8** | **100.0** | **88.5** |

### 4.3 QUANTITATIVE COMPARISON

In addition to quantitative evaluations, a qualitative analysis is conducted on the Navtest dataset to visually assess the trajectory planning performance. Figure 4 provides a comparative visualization between the proposed MTG-RPD model and the DiffusionDrive benchmark across three representative scenarios: an intersection, a straight road, and a congested area. In the figures, green dots indicate the ground-truth human-driving trajectories, while red dots depict the planned trajectories for evaluation. The first row reveals that in the intersection scenario, DiffusionDrive generates multiple trajectories that deviate beyond the drivable area, indicating a failure to comply with traffic rules. In contrast, our MTG-RPD method produces a compliant trajectory that correctly executes a right turn. The second row demonstrates the superior ego-vehicle progress achieved by our method on the straight road, highlighting its effectiveness in maintaining forward motion. The third row shows that MTG-RPD significantly reduces the planned speed in congested conditions, thereby proactively mitigating collision risks and enhancing overall safety.

## 5 CONCLUSION

In this paper, we propose MTG-RPD, a novel multimodal trajectory generation framework designed for end-to-end autonomous driving systems. The key innovation of our approach lies in the effective integration of rule-based prior trajectory into a diffusion-based generative model, which enhances the model's ability to adapt to diverse and complex driving environments while ensuring high planning accuracy and operational reliability. Extensive experiments conducted on the NAVSIM benchmark demonstrate that MTG-RPD achieves state-of-the-art performance in the PDMS and exhibits remarkable generalization across a variety of traffic scenarios. These results collectively underscore the strong practical applicability and robustness of the proposed method in real-world autonomous driving applications.

## 6 ETHICS STATEMENT

This work adheres to the ICLR Code of Ethics. This study did not involve human subjects or animal experiments. All datasets used are publicly available and do not raise privacy concerns, as they

contain no personally identifiable information. We ensured the research was conducted without bias or discriminatory outcomes and maintained the highest standards of transparency and integrity throughout the study.

## 7 REPRODUCIBILITY STATEMENT

We have made every effort to ensure the reproducibility of the results presented in this paper. All code and datasets have been made publicly available in an anonymous repository to facilitate replication and verification. The experimental setup, including training procedures, model configurations, and hardware specifications, is thoroughly documented within the paper.

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

## A APPENDIX

### A.1 LLM USAGE

Large language models were used solely to assist in the writing and polishing of the manuscript. Specifically, we utilized an LLM to help improve the language, enhance readability, and ensure clarity across various sections of the paper. The model assisted with tasks such as rewriting sentences, checking grammar, and improving the overall fluency of the text.

