# OpenReview forum: "MTG-RPD: Multimodal Trajectories Generation with Rule-Based Prior Diffusion for End-to-End Autonomous Driving"
_ICLR.cc/2026/Conference — Submitted to ICLR 2026_

### Official Review · Reviewer_Ka77 · 2025-10-30

**Soundness:** 2
**Presentation:** 2
**Contribution:** 1
**Rating:** 2
**Confidence:** 4

**Summary:**

This paper aims to design a diffusion-based framework for autonomous driving trajectory predictions to generate multimodal trajectories and improve trajectory planning by integrating rule-based prior knowledge. This framework is developed based on DiffusionDrive, which already proposed k-mean clustering to obtain anchored gaussian distributions. In the experiment, it shows that it achieves a slightly better results, though the proposed method is an extension to DiffusionDrive with rebalanced priors rather than introducing a fundamentally new diffusion framework.

**Strengths:**

1. The whole structure of the proposed method is based on DiffusionDrive and made some modifications, and this gives a better performance shown in the experiment.

**Weaknesses:**

1. It is better to give a definition to what the anchored Gaussian distributions and anchor sampling refer to. They are not common terminologies to me until I went through DiffusionDrive paper.
2. Authors claim that they design a diffusion model that supports interactions between agents and between agents and the environment. I do not quite see the novelty from this. This is a standard conditional diffusion step and it has been done in DiffusionDrive too. That is how the backbone of this proposed method is borrowed from DiffusionDrive. The only difference thing I can spot is the sparse deformable attention, and this is from deformable DETR [1], which is not cited by the paper.
3. For the rule-based prior trajectory anchor generation, the paper only briefly talks about before doing k-mean clustering, they rebalance the dataset spatially by downsampling the significance of trajectories from higher-frequented regions and elevating the lower-frequented regions. No detail is given about how to perform this and the time cost. No drawback is discussed about the rebalance: this might lead to distribution shift between anchors and real driving data and the real patterns might be overrepresented. It could also result in a misalignment between training and test-time distribution as you will not be able to rebalance the test-time distribution.
4. Samples that show the proposed method are very limited. Even though the table 1 reports the results from the other methods, but the visualizations are only on DiffusionDrive to compare.

[1] Zhu X, Su W, Lu L, Li B, Wang X, Dai J. Deformable detr: Deformable transformers for end-to-end object detection. arXiv preprint arXiv:2010.04159. 2020 Oct 8.

What I can see from this paper is the proposed method essentially builds on DiffusionDrive's diffusion trajectory planner, and it adds a rule-based prior rebalancing to accommodate low frequency traffic patterns, which achieves a slightly better performance.

**Questions:**

1. In figure 4 the third column, which shows the results from MTG-RPD, from my understanding, the orange linked dots as planned trajectories for different scenarios show very different behaviours compared to the ground truth represented by green linked dots. The vehicle in the first scene is going straight and appears going off-road, which is supposed to take a proper right turn. Same happens in the other two scenes, where the planned trajectories are not aligned with lanes. In my view, the results might be a bit better than DiffusionDrive, but it is not realistic enough to show MTG-RPD really works.
2. any discussions about the issues resulted from rebalance from the weaknesses I mentioned?

---

### Official Review · Reviewer_hhzc · 2025-10-31

**Soundness:** 3
**Presentation:** 3
**Contribution:** 2
**Rating:** 4
**Confidence:** 4

**Summary:**

This paper introduces MTG-RPD, an innovative trajectory generation method that integrates rule-based prior knowledge for end-to-end autonoumous driving.

**Strengths:**

1. The proposed hybrid approach by first getting rule-based anchor points is well motivated, which shows promising results for improving the speed.

2. Extensive results demonstrate the superiority of the proposed methods on the NAVSIM dataset.

**Weaknesses:**

1. The authors claim that the proposed method can improve trajectory planning real-time performance, but no metrics like FPS are reported against the baselines.

2. Lack of ablations. Only the usage of rule-based prior trajectory anchors is ablated. If there are more newly proposed techniques, they should be ablated as well.

**Questions:**

1. Typos: Line 83, "between agents and between agents"
2. The anonymous github link is empty, only "README.md" is in it.

---

### Official Review · Reviewer_Ez5J · 2025-10-31

**Soundness:** 2
**Presentation:** 2
**Contribution:** 1
**Rating:** 2
**Confidence:** 5

**Summary:**

The paper presents a novel approach to trajectory generation for autonomous driving by integrating rule-based prior knowledge into a conditional diffusion model. The authors claim that their method enhances the diversity and safety of generated trajectories while achieving a high performance metric (PDMS of 88.5) on the NAVSIM dataset. The approach involves generating trajectory anchor points through rule-based clustering and transforming these anchors into multimodal trajectory distributions guided by scene information.

**Strengths:**

1. **Integration of Rule-Based Knowledge**: The incorporation of rule-based prior knowledge into the diffusion model proves to be useful which leverages expert knowledge for trajectory planning, potentially improving the model's adherence to traffic regulations.

2. **Use of Diffusion Models**: The application of diffusion models in trajectory generation is timely and relevant, given the recent advancements in generative modeling techniques.

3. **Performance Metrics**: The reported performance on the NAVSIM dataset demonstrates the potential effectiveness of the proposed method in generating multimodal trajectories.

**Weaknesses:**

1. **Limited Novelty / Unclear Differentiation**: The approach appears to recombine existing techniques (rule-based priors, conditional diffusion, and cross-attention) without introducing a distinct new mechanism or learning objective. This raises questions about the novelty of the contribution, as it resembles a simple combination of known methods rather than a groundbreaking advancement.

2. **Narrow Evaluation & Incomplete Metrics**: The evaluation is limited to a single dataset (NAVSIM) and does not include other relevant datasets such as NuScenes or Bench2Drive. Additionally, the omission of the DDC sub-metric from the PDMS evaluation limits the comprehensiveness of the results.

3. **Lacking Deep Analysis**: The paper does not provide a thorough analysis of why the proposed method outperforms existing methods. The architecture used is common, and the rule-based strategy appears to function primarily as a preprocessing step rather than a core innovation.

4. **Reproducibility Issues**: The link provided for code and supplementary materials is broken, which hinders the ability to verify the results and reproduce the experiments. This is a significant concern for the credibility of the research.

**Questions:**

**Insufficient Ablations and Attribution**: The ablation studies are minimal, with only a comparison against a "w/o prior" baseline. There is a lack of systematic exploration of key parameters such as anchor scale, diffusion steps, interaction depth, and loss weights. Furthermore, the absence of a speed-vs-quality trade-off analysis raises concerns about the efficiency of the chosen diffusion steps.

---

### Meta-Review · Area_Chair_Ag5q · 2026-01-06

**Summary:**

The main concerns influencing my recommended decision are:

- Limited novelty / unclear differentiation. The method is quite similar to an existing method, "DiffusionDrive", and appears to be a straightforward combination of existing methods.
- Narrow evaluation on a single dataset. Additional popular public benchmarks could have been used to assess the method, but were not.
- Unvalidated real-time performance claim. No metrics measure the method's real-time performance.
- Lack of ablations.
- Missing details about the method

**Reviewer Concerns:**

The authors provided no rebuttal, which leaves the issues mentioned by the reviewers outstanding.

**Reviewer Scores:**

Because the authors provided no rebuttal, I do not think the reviewers would have changed their scores.

---

### Decision · Program_Chairs · 2026-01-26

Reject